# Arbitrarily routed mode-division multiplexed photonic circuits for dense integration

Yingjie Liu[1], Ke Xu[1], Shuai Wang[1], Weihong Shen[2], Hucheng Xie[1], Yujie Wang[1], Shumin Xiao[1,3], Yong Yao[1], Jiangbing Du[2], Zuyuan He[2] & Qinghai Song[1,3]

On-chip integrated mode-division multiplexing (MDM) is an emerging technique for large-capacity data communications. In the past few years, while several configurations have been developed to realize on-chip MDM circuits, their practical applications are significantly hindered by the large footprint and inter-mode cross talk. Most importantly, the high-speed MDM signal transmission in an arbitrarily routed circuit is still absent. Herein, we demonstrate the MDM circuits based on digitized meta-structures which have extremely compact footprints. 112 Gbit/s signals encoded on each mode are arbitrarily routed through the circuits consisting of many sharp bends and compact crossings with a bit error rate under forward error correction limit. This will significantly improve the integration density and benefit various on-chip multimode optical systems.

[1] State Key Laboratory on Tunable laser Technology, Ministry of Industry and Information Technology Key Lab of Micro-Nano Optoelectronic Information System, Harbin Institute of Technology (Shenzhen), 518055 Shenzhen, P. R. China. [2] State Key Laboratory of Advanced Optical Communication Systems and Networks, Shanghai Jiao Tong University, 200240 Shanghai, P. R. China. [3] Collaborative Innovation Center of Extreme Optics, Shanxi University, 030006 Taiyuan, P. R. China. Correspondence and requests for materials should be addressed to K.X. (email: kxu@hit.edu.cn) or to Y.Y. (email: yaoyong@hit.edu.cn) or to J.D. (email: dujiangbing@sjtu.edu.cn) or to Q.S. (email: qinghai.song@hit.edu.cn)

The orthogonal guiding modes in integrated optical waveguides are considered as important degree of freedom to provide the flexibility and scalability for a wide scope of on-chip applications, such as optical interconnect[1–4], quantum information science[5,6], nonlinear optics[7,8], and so on. Recently, mode-division multiplexing (MDM) is emerging as a powerful technique to boost the bandwidth of the photonic chip. Compared with wavelength division multiplexing, MDM only needs monochromatic light source, and precise wavelength control is not necessary[9]. A typical optical system requires a variety of multimode devices such as (de)multiplexer[10–12], grating coupler[13], switch[14–16], mode filter[17], splitter[18,19], and many other building blocks[20–22]. The interconnections among these elements have to rely on multimode waveguides with a few orthogonal and co-propagating modes. This is quite challenging, since the signal transmission in such multimode photonic circuits is vulnerable to sharp bending and cross connection due to radiation leakage and inter-mode coupling. To achieve adiabatic wave propagation in a bent waveguide, the required footprint is too large[23]. The mode-independent crossing is also complicated for higher-order modes since the MDM signals needs to be demultiplexed to single modes prior to waveguide crossing[24]. For both cases, the MDM signal is routed at the cost of large chip area, which eventually limits the integration density.

To arbitrarily route the multimode waveguide, the bending and crossing are two key elements. Recently, various waveguide structures have been proposed to achieve low-loss and low-inter-mode-crosstalk (CT) waveguiding for the bend and cross connection scenario[25–31]. However, most of the demonstrations only achieve dual-mode bending and crossing. Recently, Gabrielli et al. have shown a three-mode bending with low inter-mode coupling, but the bending radius is large and gray-scale lithography is needed[29]. A three-mode crossing has also been reported based on Maxwell fish eye lens structure with a footprint of $11.6 \times 11.6\ \mu m^2$ where a precisely designed focusing taper is required[31].

In this article, we demonstrate ultra-compact and arbitrarily routed circuits for three modes via the unprecedentedly small bending (3.9-μm radius) and crossing ($8 \times 8\ \mu m^2$). The sharp bending and direct cross connection for $TE_0$, $TE_1$, and $TE_2$ modes are achieved by the sophisticated waveguiding, which is realized by a discretized meta-structure with a number of nanoholes. Optimization algorithm is utilized to design the complex structure of nanohole distribution. The tailored index profile allows for mode matching between the multimode waveguide and the bending/crossing region. Furthermore, our proposed devices can be compatible with standard silicon photonics foundry fabrication process (see Supplementary Note 5). We have achieved a three-mode bending with 0.71–0.95 dB loss and a crossing with 0.28–0.82 dB loss over 80 nm bandwidth. Finally, 112 Gbit/s signals encoded on each mode are successfully routed along the arbitrarily designed circuits with a bit error rate (BER) under forward error correction (FEC) limit.

## Results

### Three-mode bending.
The micro-bend with a digital meta-structure (DMS) is designed and illustrated in Fig. 1a. The bending region is divided into discrete pixels with minimum feature size of 130 nm. The device platform is silicon-on-insulator (SOI) with 220 nm top silicon and 2 μm buried oxide. We define a binary material state for each pixel, i.e., it can be either silicon or air (by etching away the silicon). This allows for a local index contrast of $\Delta n = 2.48$ to manipulate the wave propagation. The waveguide width is chosen to be 2.3 μm, which allows for enough number of pixels. The nanohole (air pixel) distributions can be determined by various optimization algorithms[32–37]. Here we use

a direct binary search (DBS) algorithm to optimize the structure (see "Methods"). First, we have designed an extremely compact bend with an inner curvature radius of only 2.75 μm (effective radius: 3.9 μm) for a three-mode division multiplexed system. The bend can support transmission of $TE_0$, $TE_1$, and $TE_2$ mode with low-loss and low-inter-mode CT. The optical field distributions for different modes are simulated by three-dimensional finite difference time domain (FDTD) method and are shown in Fig. 1b–d. It can be seen that the optical waves for all the modes are squeezed and focused to the corner region of the bend by a few nanoholes near the inner sidewall of the bent waveguide. The outer bound nearly reflects the beam to the output waveguide with preserved mode profile. Though this is not an absolute adiabatic process, the inter-mode interference can be well suppressed to a negligible level. We also quantitatively study how these nanoholes affect the device performance by filling some of the holes with silicon and re-simulate the transmission spectra. We found that the three isolated holes near the outer curvature had nearly negligible impact on the transmission of all the modes. The efficiency drops <2% without these air holes. However, the other nanoholes are critical to the device performance. The simulated spectra of the device from 1500 to 1580 nm are shown in Fig. 1e–g. The simulated insertion losses (ILs) are 0.4, 0.7, and 0.84 dB for $TE_0$, $TE_1$, and $TE_2$ modes, respectively. The CTs are $<-20$ dB for all the modes over the 80-nm wavelength range, which is sufficient for negligible signal-to-noise ratio (SNR) penalties[2,4].

The device fabrication process can be found in "Methods". To demonstrate the sharp bending of a few-mode waveguide, we fabricated an MDM circuit consisting of a mode multiplexer (MUX), two cascaded bends, and a mode demultiplexer (DEMUX), as shown in Fig. 2a. The MUX is also a compact DMS with a footprint of only $3.4 \times 3.9\ \mu m^2$, which is described in details in Supplementary Note 1. As illustrated in Fig. 2a, the circuit has three input ports labeled as I1 ($TE_0$ mode), I2 ($TE_1$ mode), and l3 ($TE_2$ mode) and three output ports labeled as O1–O3 ($TE_0$-$TE_2$ modes), respectively. In the meantime, as shown in Fig. 2b, an extra reference circuit with a back-to-back mode MUX is fabricated to normalize the transmission of the device and to extract the bending loss. The transverse electric (TE)-polarized grating coupler is used to interface the fiber and the waveguide. The coupling loss is ~6 dB per facet. The DMS-based bent waveguide is well fabricated as we designed, which can be seen from the zoom-in scanning electron microscope (SEM) image shown in Fig. 2c. The measurement set-up for the transmission spectra of the device are described in "Methods". The normalized transmission spectra measured at O1–O3 ports are plotted in Fig. 2d–f when the continuous wave is launched from input ports I1–I3, respectively. The bending loss is measured to be 0.71, 0.74, and 0.95 dB for $TE_0$, $TE_1$, and $TE_2$ modes, respectively. The measured CTs are $<-20$ dB for all the guiding modes from 1500 to 1580 nm, which is quite consistent with the simulation results.

### Three-mode crossing.
The on-chip routing inevitably needs to address the waveguide cross-connect problem. Here we design a three-mode crossing with a footprint of only $8 \times 8\ \mu m^2$. The structural schematic diagram of the device is described in Fig. 3a. The device has a quadrature symmetry with the minimum feature size of 130 nm. The input/output waveguide widths of the crossing are set to be 1.4 μm to support $TE_0$, $TE_1$, and $TE_2$ modes. Each quadrature area is axial symmetric and occupies a chip area of only $4 \times 4\ \mu m^2$. This design region is divided into $25 \times 25$ discrete pixels. The optimization process is similar to the design of bending. Figure 3b–d show the simulated in-plane optical field profiles of the nanostructured crossing when the input waves are

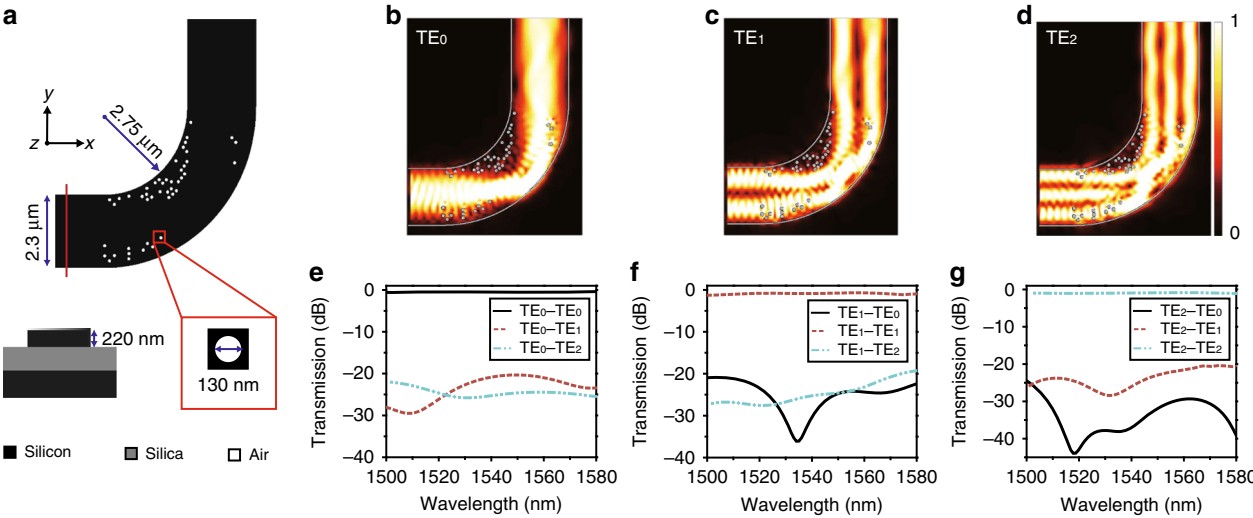

**Fig. 1** Design and simulation results of the digital meta-structure (DMS)-based bending. **a** Schematic diagram of the designed three-mode bending with DMS. **b–d** The simulated optical field distributions for the micro-bend with three modes (TE$_0$–TE$_2$). **e–g** The simulated transmission spectra for the micro-bend with three modes (TE$_0$–TE$_2$)

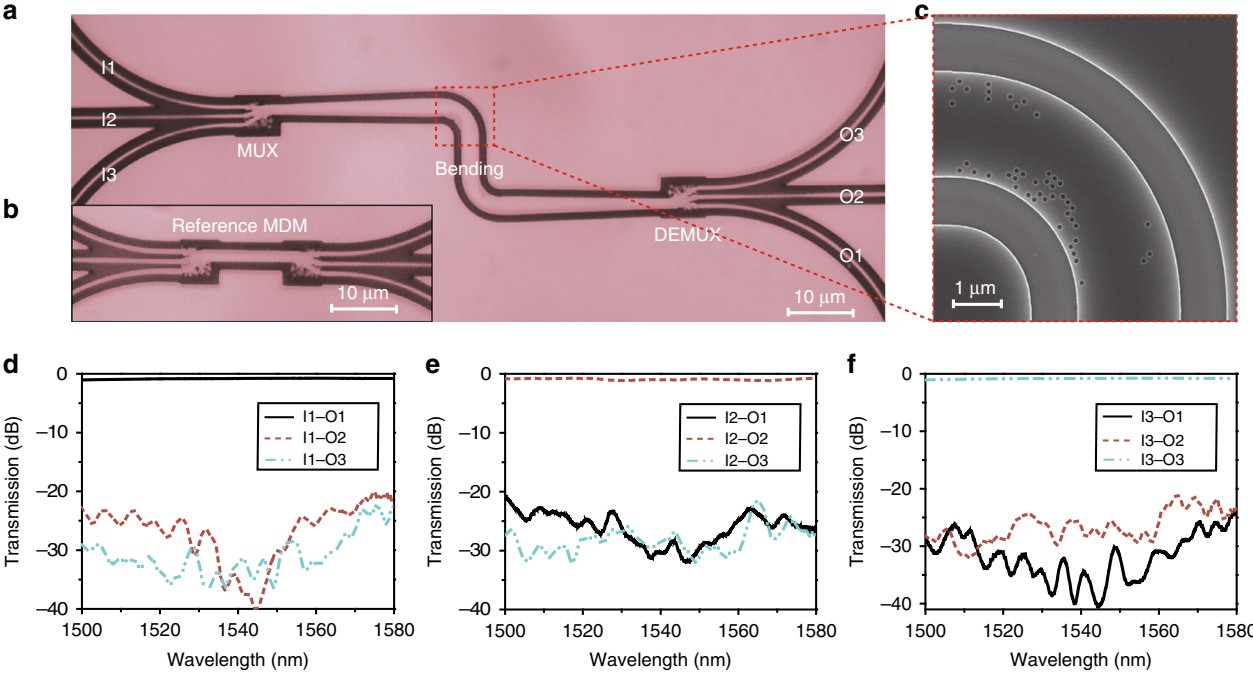

**Fig. 2** Fabrication and experimental results of the digital meta-structure (DMS)-based bending. **a** The top-view microscope image of the mode-division multiplexing (MDM) circuit consisting of multiplexer (MUX), bending, and demultiplexer (DEMUX). **b** The microscope image of the reference circuit: back to back (DE)MUXs. **c** The scanning electron microscope image of the fabricated bending. **d–f** The measured transmission spectra of the MDM circuit for the optical wave launched from I1 (TE$_0$ mode), I2 (TE$_1$ mode), and I3 (TE$_2$ mode), respectively

TE$_0$, TE$_1$, and TE$_2$ modes, respectively. There are several nano-holes located at the corner region of the crossing, which provides a forbidden gap with low index to avoid mode expansion. Hence, the mode profile of each mode can be well preserved when propagating through the crossing region. There is a small amount of mode leakage into the orthogonal waveguide. This can be further reduced by setting an additional optimization objective to reduce such leaked power. To quantitatively investigate the performance of the crossing, we calculate the transmission efficiency and the mode CT taking mode overlap integral into account.

Figure 3e–g show the calculated IL and CT spectra. The simulated ILs for TE$_0$, TE$_1$, and TE$_2$ modes are 0.2, 0.55, and 0.59 dB, respectively. The simulated CTs are <−30 dB for all modes from 1500 to 1580 nm wavelength range.

We experimentally characterize the crossing performance by fabricating the three-mode cross-connect structure. The microscope image of the circuit is shown in Fig. 4a. The devices were fabricated using the same process as the bending. Owing to the ultra-low loss of an individual crossing, we cascade different numbers of crossing to accurately extract the IL of the device.

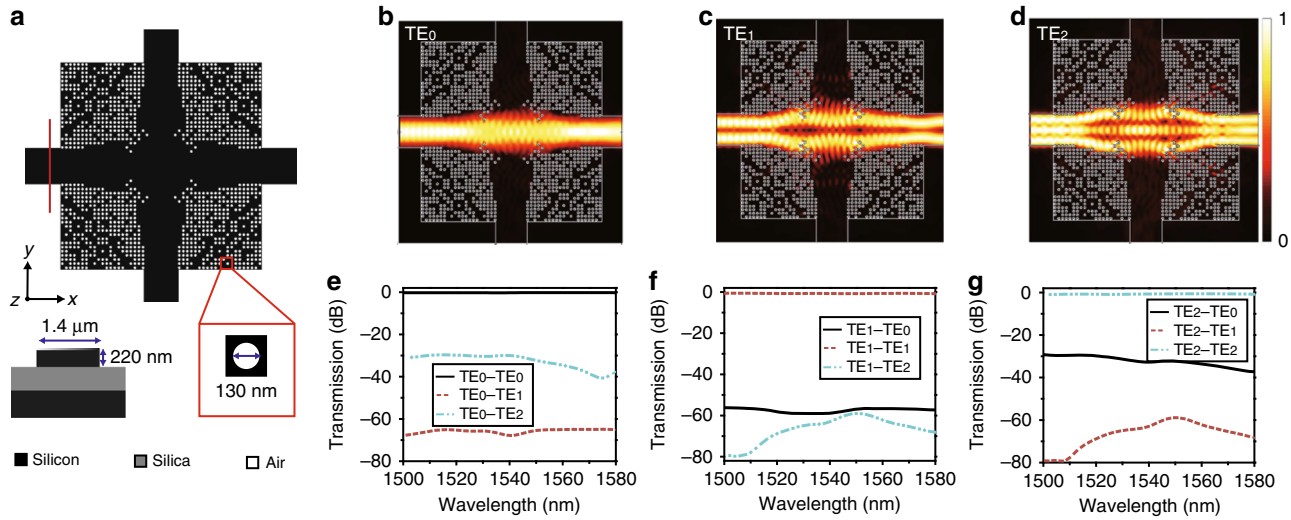

**Fig. 3** Design and simulation results of the three-mode crossing. **a** Schematic diagram of the designed three-mode crossing with digital meta-structure. **b–d** The simulated optical field distributions for the three-mode crossing with $TE_0$–$TE_2$. **e–g** The simulated transmission spectra for the three-mode crossing with $TE_0$–$TE_2$

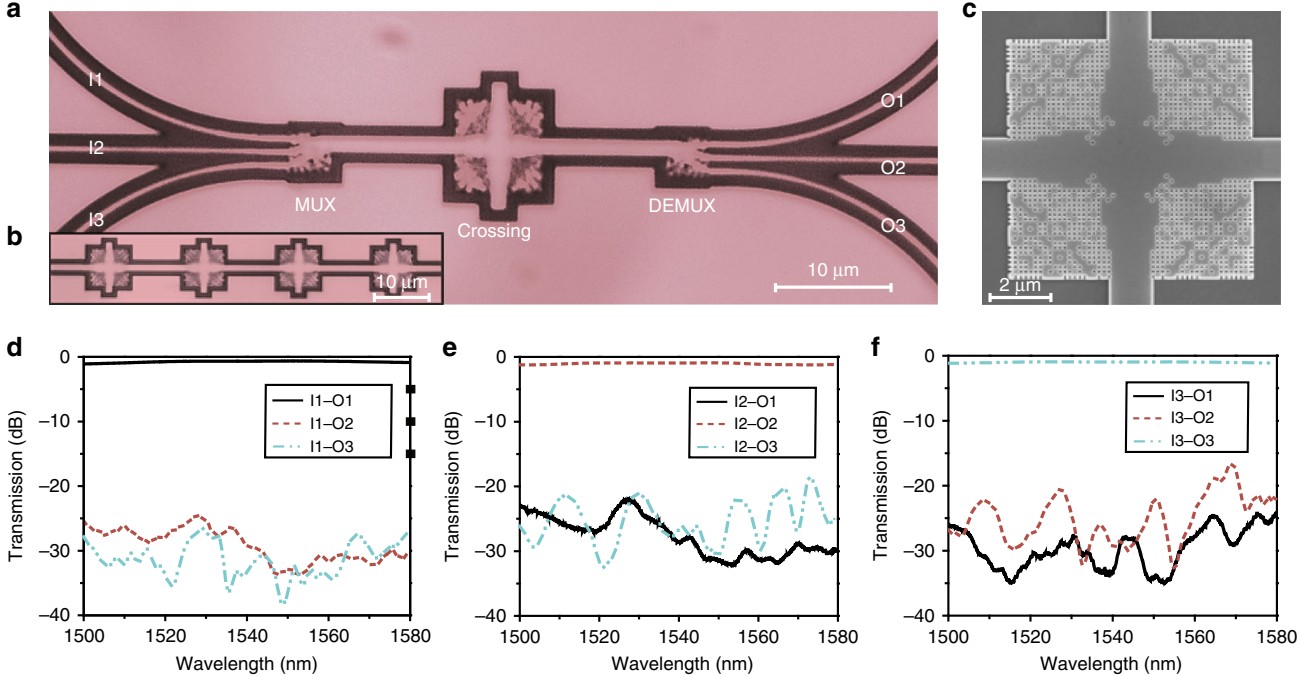

**Fig. 4** Fabrication and experimental results of the three-mode crossing. **a** The top-view microscope image of the mode-division multiplexing (MDM) circuit consisting of a multiplexer, a crossing, and a demultiplexer. **b** The microscope image of the four-cascaded crossing. **c** The scanning electron microscope image of the fabricated crossing. **d–f** The measured transmission spectra for the MDM circuit

Figure 4b shows the microscope image of a four-stage cascaded crossing circuit, which is more measureable than a single crossing. In Fig. 4c, the zoom-in SEM image of the nanos-tructured crossing confirms that the desired pattern has been successfully transferred to the device layer. The transmission spectra of the cascaded crossing are measured and normalized. The measured ILs and CTs for $TE_0$-$TE_2$ modes of a single crossing are shown in Fig. 4d–f, respectively. The ILs are 0.28, 0.68, and 0.82 dB for $TE_0$, $TE_1$, and $TE_2$ modes, respectively. The CTs for all modes are characterized by measuring the optical power at the other output ports. The spectra indicate that the CTs are <−20 dB for all the three guiding modes.

**MDM routing circuits.** Most of the previous demonstrations on MDM circuit hardly routed the multimode waveguides due to the limited capability of sharp turning and compact cross connection. Here we demonstrate the possibility of arbitrary and compact on-chip routing of high-speed signals (112 Gbit/s) in MDM circuits. In principle, the MDM signal can be delivered to any chip loca-tions via any routing path provided the low loss bending and crossing are available. For a proof-of-concept demonstration, we arbitrarily design two MDM circuits with the proposed bending and crossing. MDM circuit 1 consists of a MUX, four bends, a crossing, and a DEMUX. The microscope image of the three-mode division multiplexed circuit is shown in Fig. 5a. Such circuit

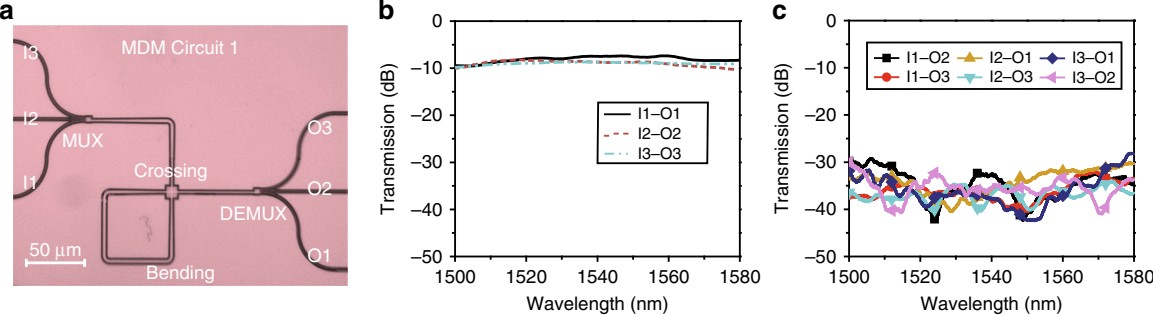

**Fig. 5** Fabrication and experimental results of the mode-division multiplexing (MDM) circuit 1. **a** The top-view microscope image of the MDM circuit 1 consists of a multiplexer, 4 bends, a crossing, and a demultiplexer. **b** The measured insertion losses for different input/output ports. **c** The measured crosstalk performances for different input/output ports

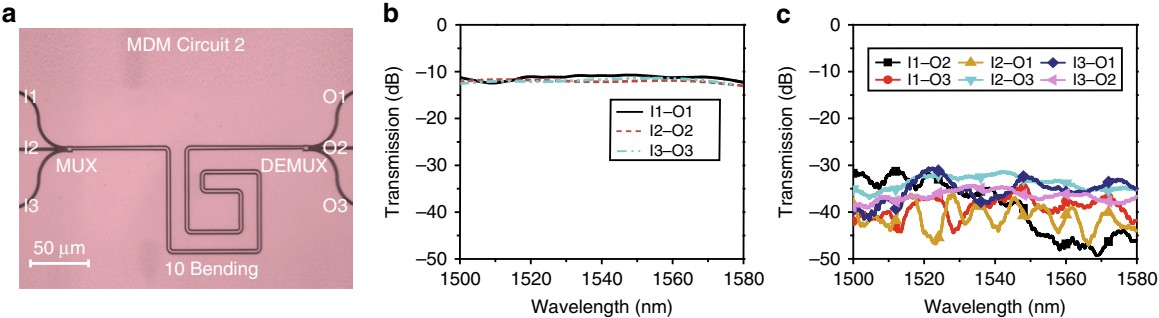

**Fig. 6** Fabrication and experimental demonstration of the mode-division multiplexing (MDM) circuit 2. **a** The top-view microscope image of the MDM circuit 2 consists of a multiplexer, 10 bends, and a demultiplexer. **b** The measured insertion losses for different input/output ports. **c** The measured crosstalk performances for different input/output ports

allows both sharp bending and cross connection, which can be scaled to cascaded structure or waveguide array scenarios. We define input ports I1–I3 and output ports O1–O3 for $TE_0$-$TE_2$ modes, respectively. Fig. 5b, c show the measured IL and CT spectra for all the modes after transmission through the circuit. The losses come from the MUX, DEMUX, two times cross connection, and four bending. The average IL is ~8 dB and the CTs are $<-20$ dB within the wavelength range from 1500-1580 nm.

MDM circuit 2 consists of a MUX, ten bends, and a DEMUX. The waveguide is configured in a spiral route and can be further scaled to a dense spiral waveguide. This circuit is used to confirm the ability to continuously change the routing direction. The microscope image of MDM circuit 2 is shown in Fig. 6a. The measured IL and CT spectra of circuit 2 are shown in Fig. 6b, c. The average IL for all the mode channels is 12 dB from 1500 to 1580 nm. The CT of MDM circuit 2 is measured to be $<-20$ dB within the same wavelength range.

In addition to simply measuring the circuit loss and CT, we verified the capability of high-speed signal routing via the proposed MDM circuits. Each stream of 112 Gbit/s discrete multi-tone (DMT) signals with identical bit allocation are transmitted through each channel of the MDM circuits one by one. Though we measure each mode channel of signal transmission separately, we would expect the potential of simultaneous transmission of the mode-multiplexed signals. This is because the additional signal penalties mainly come from the inter-mode CT of the MDM circuits. According to the theoretical predictions[2,4], the additional penalties induced by CT are insignificant for the demonstrated routing circuits. The detailed experimental set-up and the digital signal processing technique

can be found in Supplementary Note 4. The measured SNR responses at the back-to-back (B2B) case and MDM transmission are plotted in Fig. 7a. The SNR responses are roughly unchanged after the MDM circuit transmission with very slight difference compared with B2B case, and the differences imply the diverse transmission performance among the three channels due to the IL, CT, and even the circumstance instability. The SNR response measured at the B2B case is used to generate a DMT signal by bit-allocation algorithm, shown in Fig. 7b. The maximum bit allocation is 5 which corresponds to 32-QAM. The same DMT signals are then routed in different mode channels one by one. The spectrally efficient DMT technique allows us to achieve 112 Gbit/s single lane rate under a bandwidth-constrained condition. The constellations with a bit index of 5 (32-QAM) and 4 (16-QAM) under different subcarrier bands are plotted as the inset of Fig. 7b, which indicates a good signal quality for high-order modulation formats. BER measurement is performed, and the results are shown in Fig. 7b. The measured BER curves of 112-Gbit/s DMT are plotted in Fig. 7c, which are mostly below the 20% FEC limit[38]. Thus the ultra-compact signal routing of high-speed signals is successfully achieved. Besides, the characterization has also been performed at lower speed of 80-Gbit/s 4-level pulse-amplitude modulation (PAM-4) (See Supplementary Note 4).

## Discussion

We demonstrate the MDM circuit that can be arbitrarily routed using extremely compact bending and crossing. The performances of the devices are summarized in Supplementary Note 2.

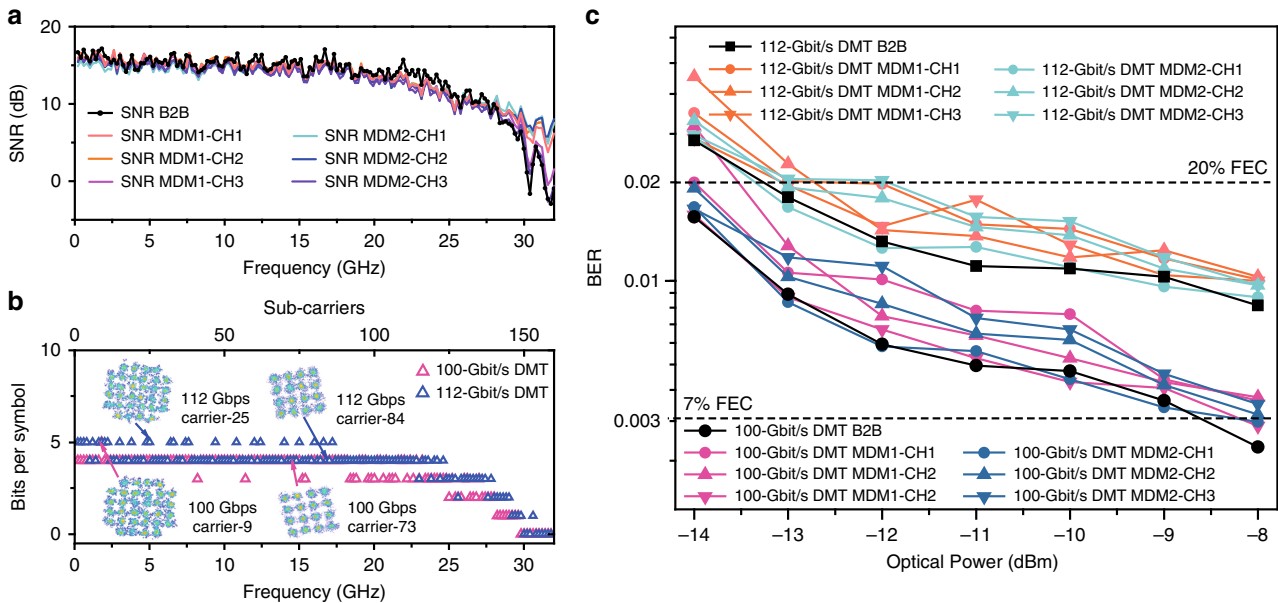

**Fig. 7** Experimental results of the high-speed signal routing in mode-division multiplexing (MDM) circuits. **a** The signal-to-noise ratio response for back-to-back (B2B) and MDM circuits transmission. **b** The bit allocation for different subcarriers. Inset: constellations of different subcarriers. **c** Measured bit error rate curves for the B2B and MDM transmission of 100 and 112 Gbit/s discrete multi-tone

Manipulating the optical wave in such a tiny structure relies on the index engineering in a deep subwavelength scale. An optimized index profile is the key to mode matching, which helps us to avoid mode leakage and inter-mode coupling. The index contrast of the nanoholes with air cladding is $n = 2.48$. However, a thick oxide cladding is normally deposited on the silicon layer, which reduces the index contrast by 20%. Here we have confirmed in the simulations that the oxide cladding does not affect the design of such compact MDM device. The digitized meta-structure for bending and crossing with oxide cladding are designed following IMEC's design rule (see Supplementary Note 5).

It is important to verify the contributions from those discretized nanoholes. We have numerically investigated the performance of the bending and crossing structure if all the etched pixels are filled by silicon again. For such a sharp bend and compact crossing waveguide, the ILs and CTs increases significantly as we expect (See Supplementary Note 6). This in turn proves the evidence of the effectiveness of index engineering via inverse design.

## Methods

**Device design.** First, discretization is applied to the design area of the device. The geometry of each pixel can be circle, square, or other complex shapes. The minimum feature size can be determined according to the fabrication capability. Here each pixel can be either high refractive index dielectric or low index cladding. The DBS method is used to find a proper distribution of the binary pixels that meets the design targets. For the initial pattern of the device, all the pixel states are chosen to be silicon. Then the material state of each pixel is altered one by one, and the figure-of-merits (FOMs) are inspected. Here the FOMs are ILs and CTs for TE$_0$-TE$_2$ modes, which are calculated by 3D FDTD with 30 nm × 30 nm × 30 nm grid size. If the FOMs improve, the pixel state alteration will be effective and be saved. The improvement can be verified by the following condition:

$$\left[ \frac{\sum_j \eta_{j,i+1}}{\sum_j \eta_{j,i}} \geq 1 \right] \cap_{j=2}^{3} \left[ \left| \frac{\eta_{j,i}}{\eta_{1,i}} - 1 \right| \leq \sigma_{j,i} \right] \quad (1)$$

and

$$\left[ \frac{\sum_k \beta_{k,i+1}}{\sum_k \beta_{k,i}} \leq 1 \right] \cap_{k=2}^{6} \left[ \left| \frac{\beta_{k,i}}{\beta_{1,i}} - 1 \right| \leq \rho_{k,i} \right] \quad (2)$$

where $\eta_{j,i}$ is the transmission efficiency of the $j$th mode from 1500 to 1580 nm in the $i$th iteration ($j = 1$ for TE$_0$, $j = 2$ for TE$_1$, and $j = 3$ for TE$_2$), and $\beta_{k,i}$ is the $k$th CT from 1500 to 1580 nm in $i$th iteration ($k = 1$ for TE$_0$-TE$_1$, $k = 2$ for TE$_0$-TE$_2$, $k = 3$ for TE$_1$-TE$_0$, $k = 4$ for TE$_1$-TE$_2$, $k = 5$ for TE$_2$-TE$_0$, and $k = 6$ for TE$_2$-TE$_1$). $\sigma_{j,i}$ is the radius of convergence for efficiency of the $j$th mode in the $i$th iteration, and $\rho_{j,i}$ is the radius of convergence for the $k$th CT in the $i$th iteration, which is a monotone decreasing function.

One iteration ends after all the pixel states are inspected. Then this process goes over again to further improve the FOMs to design objectives. It takes several iterations to meet the condition of convergence where $\sigma$ and $\rho$ converge to zero:[39]

$$\sigma_{j,i} = \varepsilon \times \left( 1 - \frac{\eta_{j,i}}{\eta_{j,\text{objective}}} \right) \quad (3)$$

and

$$\rho_{k,i} = \delta \times \left( \frac{\beta_{k,i}}{\beta_{k,\text{objective}}} - 1 \right) \quad (4)$$

where $\varepsilon$ and $\delta$ are the convergence factor, $\eta_{j,\text{objective}}$ is the transmission objective of the $j$th mode, and $\beta_{k,\text{objective}}$ is the transmission objective of the $k$th CT.

We perform the device design and optimization using an eight-core desktop. For the bending device, it takes ~50 h on average to get the convergent results after 4 iterations. For the crossing device, it takes ~40 h after 3 iterations.

**Device fabrication process.** The devices are fabricated on a SOI wafer with 220 nm top silicon device layer on 2-μm buried oxide. The positive photo resist ZEP 520A is used as the soft mask on SOI. The devices are patterned by the electron beam lithography (Raith eLINE), which operates at 30 kV, then the top silicon layer was fully etched to a depth of 220 nm by using a single-step inductively coupled plasma dry etching.

**Measurement set-up and methods.** For device characterization, the set-up consists of a sweep laser (Yenista T100S-HP/SCL), a fiber-chip coupling stage, and a benchtop power meter. The continuous wave output of the laser is TE polarized via a polarization controller. Grating couplers are used to interface the single-mode fibers and the silicon waveguides. A tilt angle of 10 degrees with respect to the surface normal direction is designed to suppress the back-reflection. The coupling loss of each grating coupler is measured to be ~6 dB. The transmission spectra are obtained by measuring the chip output via a benchtop power meter. For high-speed experiments, the set-up is described in Supplementary Note 4.

## Data availability

The data that support the findings of this study are available from the corresponding authors upon reasonable request.

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

## Acknowledgements

This work is supported by National Natural Science Foundation of China (NSFC) (Grant Nos. 61875049, 61675128, and 61875124) and Shenzhen Science and Technology Innovation Commission (Grant Nos. JCYJ20170307151047646, JCYJ20180507183418012, and KQJSCX20180328165451777).

## Author contributions

K.X. conceived the project. Y.L. and H.X. performed the numerical simulations. S.W., Y.W., and Y.L. fabricated the MDM devices. Y.L. characterized the MDM components and circuits. W.S. and J.D. performed the high-speed experiments. Y.L., K.X., J.D., S.X., Y.Y., Q.S., and Z.H. discussed and analyzed the measured data. Y.L. and K.X. wrote the first draft of the manuscript. All the authors discussed the results and contributed to the writing of the paper.

## Additional information

**Competing interests:** The authors declare no competing interests.

