## [Peer Review File · Nature Communications]

Reviewers' comments:

Reviewer #1 (Remarks to the Author):

The manuscript by Liu et al. describes a silicon photonic circuit for mode multiplexing. The waveguides contain nano-hole features that improved the performance of the device. The work is acceptable, though it is unclear to me whether the work would interest a broad readership. There has been other works on mode multiplexing silicon photonic circuits by D. Dai (Zhejiang University), as well as M. Lipson and Bergman (Columbia University). However, different types of photonic circuits were demonstrated. The submission would definitely be suitable to a technical field journal, such as Optics Express, Optics Letters, and perhaps Optica.

Here are some technical comments for this manuscript:

1. The Si photonic chip was fabricated using electron-beam lithography and did not have a cladding. This is not considered to be CMOS compatible. Although the minimum feature size is kept to 130nm, it is well known that e-beam lithography will produce better defined features than a photolithography process. The minimum feature size requirement in a foundry is usually referring to minimum line spacing or widths, and does not guarantee the fabrication of small hole-like features. Therefore, this work so far cannot be claimed to be CMOS compatible or Si photonic foundry compatible.
2. The light was coupled into/out of the chip using grating couplers that coupled to the TE₀ mode on chip. The high order modes were generated and demultiplexed/multiplexed on chip using the MMI. How can the authors be sure that the TE₁ and TE₂ modes were generated on chip?
3. The crosstalk of the bend was limited to about ~-15dB (Fig. 2(d)). That is not very low. Usual telecom requirements demand < -30dB. What are some ways to reduce the cross-talk?
4. What is the fabrication tolerance of the devices? How sensitive is the device performance to the size and position of the holes?

Reviewer #2 (Remarks to the Author):

The paper summarizes results of an interesting inverse design approach to realizing key passive components for routing 3-mode waveguides in Si-photonic integrated circuits.

The paper is generally well written and might become suitable for publication provided a few key comments are addressed:

1. The paper is generally well structured but would benefit a lot from the addition of a table summarizing the performance (loss, cross talk etc.) of the different components.
2. The use of the DMT format to characterize the devices seems quite unfortunate. As I understand the explanation in the paper, the bit loading is optimized in each of the cases (B2B, MDM1, MDM2). This makes sense if the purpose was to maximize the amount of transmitted information. However, given that the purpose here is to characterize the devices it would have been far better to use exactly the same signal in all cases. As it is now one needs to ask the question: why does the optimum bit loading change? You have almost the same SNR in all cases, so what is happening in the device to cause this change?
3. Please state the exact bitrate subtracting the overhead due to FEC. Given that the bit loading changes, it would seem that the bitrate is different in the three cases.

4. As I read the description in the supplementary material, it seems like only one modulated data signal was passed through the device when characterizing it. Is this the case? If so, why does the paper refer to 3x100 Gbit/s? This should be clarified/corrected.

5. It is very relevant that the tolerance to fabrication error on the hole size in the structure is characterized, however, what about errors in placement and variation in shape? I think the authors should comment on this in the paper and justify why size variation is the most sensitive to fabrication errors.

6. Other than causing loss, fabrication errors might also cause increased crosstalk. In actual MDM communication this could be a much more severe limitation than a slight increase in loss. I think the authors should investigate the relevant fabrication tolerances, also in terms of increased crosstalk.

Reviewer #3 (Remarks to the Author):

In an ``Arbitrarily routed mode-division multiplexed photonic circuits for dense integration' Yingjie Liu et al. have presented a collection of basic (inverse designed) building blocks for use in on-chip mode division multiplexing photonic circuits. This approach presents major advantages over traditional wavelength division multiplexing, and will likely continue to grow in popularity as photonic optimization techniques mature. Although conceptually unremarkable in light of previous publications, the demonstration provided by the authors is thoroughly convincing and offers clear performance advantages over any existing art. With minor revisions to improve the quality of the text, it is entirely suitable for publication in Nature Communications.

Comments:

1. Some additional details about the direct binary search should be included. What does an iteration refer to in this context? What are the convergence conditions for each structure? The figures of merit?

2. The text has numerous small grammatical errors that need to be corrected. For instance:

The adiabatic wave propagation requires too large footprint for a bent waveguide.

The waveguide width is chosen to be 2.3 μm which can support three lowest order modes and allow for enough number of pixels as well.

DBS method is easy to implement and fast to convergence for the optimization of digital meta-structure with binary material state.

Reviewer #1 (Comments to the Author):

The manuscript by Liu et al. describes a silicon photonic circuit for mode multiplexing. The waveguides contain nano-hole features that improved the performance of the device. The work is acceptable, though it is unclear to me whether the work would interest a broad readership. There has been other works on mode multiplexing silicon photonic circuits by D. Dai (Zhejiang University), as well as M. Lipson and Bergman (Columbia University). However, different types of photonic circuits were demonstrated. The submission would definitely be suitable to a technical field journal, such as Optics Express, Optics Letters, and perhaps Optica.

Reply: We sincerely thank the reviewer for the comment. The photonic integration has achieved a great success in telecom and datacom. Growing efforts have been directed to implementing many other optical systems on a photonic chip to take the benefits of integration. In addition to wavelength, a set of orthogonal waveguide modes is another important degree of freedom in integrated optical waveguides to provide the flexibility and scalability of the on-chip functional system. The MDM for large-capacity on-chip optical interconnect is an active area where a variety of functional elements are to be exploited. More recently, on-chip multimode system attracted much interests in other applications like quantum information science (L. Feng, et al, Nat. Comm., 7, 11985, 2016; L. Feng, et al, NPJ Quantum Information, 5, 2, 2019) and nonlinear optics (E. Kittlaus, et al, Nat. Comm., 8, 15819, 2017; E. Kittlaus, et al, Nat. Photon., 12, 613, 2018). As the system becomes more complicated along with the increased functionality, the compact waveguide routing proposed in this work will benefit all the applications based on large scale integration of multimode optical system.

Revisions:

Abstract: This will significantly improve the integration density and benefit various on-chip multimode optical system.

Page1, paragraph1:

The orthogonal guiding modes in integrated optical waveguides are considered as important degree of freedom to provide the flexibility and scalability for a wide scope of on-chip applications such as optical interconnect¹⁻⁴, quantum information science^{5, 6}, nonlinear optics^{7, 8}, and so on. Recently, mode-division multiplexing (MDM) is emerging as a powerful technique to boost the bandwidth of the photonic chip. Compared with wavelength division multiplexing, MDM only needs monochromatic light source, and precise wavelength control is not necessary⁹. A typical optical system requires a variety of multimode devices such as (de)multiplexer¹⁰⁻¹², grating coupler¹³, switch¹⁴⁻¹⁶, mode filter¹⁷, splitter^{18,19} and many other building blocks²⁰⁻²².

Here are some technical comments for this manuscript:

1. The Si photonic chip was fabricated using electron-beam lithography and did not have a cladding. This is not considered to be CMOS compatible. Although the minimum feature size is kept to 130nm, it is well known that e-beam lithography will produce better defined features than a photolithography process. The minimum feature size requirement in a foundry is usually referring to minimum line spacing or widths, and does not guarantee the fabrication of small hole-like features. Therefore, this work so far cannot be claimed to be CMOS compatible or Si photonic foundry

compatible.

Reply: We thank the reviewer for raising this issue. We agree with the reviewer that the silicon photonics foundries cannot fabricate the nanoholes with such narrow gap spacing via photolithography. Many silicon photonics foundries provide 193nm UV lithography to define the device pattern. For example, IMEC silicon photonics ISIPP50G technology allows for a minimum hole size of 130 nm (diameter) and hole gap distance of 120 nm.

To demonstrate the compatibility with silicon photonics foundry fabrication, we follow IMEC's design rule (hole size: 130 nm in diameter; hole gap distance: 120 nm; oxide cladding) and re-optimize the devices. The detail of the simulation results can be found in the revised Supplement Information Note4. We can observe negligible performance degradation for the mode (de)multiplexer, waveguide bending and crossing. As a result, such algorithm-optimized digital meta-structures can be possibly compatible with the design rule given by silicon photonics foundry. Since the designed devices are fabricated using E-beam lithography for a proof-of-concept demonstration, we have rephrased the statement in the manuscript according to the reviewer's suggestion.

Revisions:

In page 1, paragraph 3, the statement is changed into "Furthermore, our proposed devices can be compatible with standard silicon photonics foundry fabrication process (see Supplementary Note5)."

[See **Supplementary Note5**]:

Supplementary Note5 | Compatibility with silicon photonic foundry fabrication

"Here, we demonstrate the devices' compatibility with silicon photonics foundry fabrication. We follow IMEC's design rule (hole size: 130 nm in diameter; nanohole gap distance: 120 nm; oxide cladding) and re-optimize the devices. Supplementary Figure 17 (a) illustrates the schematic diagram and the structural parameters of the re-designed bending with oxide cladding. The simulated optical field distribution of the bending for TE_0 , TE_1 and TE_2 at 1550 nm are shown in Supplementary Figure 17 (b) – (d), respectively. Supplementary Figure 17 (e) - (g) show the simulated transmission spectra of the bending structure for $TE_0 - TE_2$ from 1500 to 1580 nm. For $TE_0 - TE_2$, the ILs of all modes are less than 1 dB and the CTs are lower than -20 dB. Similarly, we re-design the waveguide crossing and (de)MUX following IMEC's design rule. The device schematic diagram, optical field distributions, transmission curves of the crossing and (de)MUX are shown in Supplementary Figure 18 and 19, respectively. For all the modes, the ILs are less than 1 dB and the CTs are lower than -20 dB for both devices. Hence, the MDM devices proposed in this work can be fully compatible with the design rule given by the silicon photonics foundry."

Supplementary Figure 17 | Design and simulation results of the bending following the foundry's design rule. (a) Schematic of the optimized bending following IMEC's design rule. **(b) - (d)** The simulated optical field distribution of the bending for $TE_0 - TE_2$ at 1550 nm. **(e) - (g)** The simulated transmission spectra of the bending for $TE_0 - TE_2$ from 1500 to 1580 nm.

Supplementary Figure 18 | Design and simulation results of the crossing following the foundry's design rule. (a) Schematic of the optimized crossing following IMEC's design rule. **(b) - (d)** The simulated optical field distribution of the crossing for $TE_0 - TE_2$ at 1550 nm. **(e) - (g)** The simulated transmission spectra of the crossing for $TE_0 - TE_2$ from 1500 to 1580 nm.

Supplementary Figure 19 | Design and simulation results of the MUX following the foundry's design rule. (a) Schematic of the optimized MUX following IMEC's design rule. **(b) - (d)** The simulated optical field distribution of

the (de)MUX for $TE_0 - TE_2$ at 1550 nm. (e) - (g) The simulated transmission spectra of the (de)MUX for $TE_0 - TE_2$ from 1500 to 1580 nm.

2. The light was coupled into/out of the chip using grating couplers that coupled to the TE_0 mode on chip. The high order modes were generated and demultiplexed/multiplexed on chip using the MMI. How can the authors be sure that the TE_1 and TE_2 modes were generated on chip?

Reply: We thank the reviewer for bringing our attention to this point. To confirm that the $TE_0 - TE_2$ modes were generated by the mode MUX device, we fabricated the structure as shown in Figure R1 (a). The input single mode waveguides are connected to three grating couplers (I1-I3) for signal input and the output is a cleaved edge at the end of the straight multimode waveguide (O). A mode MUX device based on digital meta-structure is used to multiplex the three fundamental modes into $TE_0 - TE_2$ into the multimode waveguide. The zoom-in SEM images of the designed mode multiplexer and multimode waveguide can be seen in Figure R1 (b) - (c).

Then, we measured the far-field profiles when the optical wave was launched from input I1-I3 to verify the mode profiles generated in the multimode waveguide. Figure R2 is the schematic diagram of experimental setup for characterizing the mode profiles generated by the mode multiplexer. The continuous wave output of a C-band tunable laser (Yenista T100S-HP/SCL) is fed into the chip via grating coupler. Then, an infrared camera (Duma Optronics, BA3-IR3E-USB) was used to capture the optical far field profile of the emission from the end of output waveguide (O). Figure R3 (a) - (c) show that the measured far-field of output when the light wave is launched from I1-I3, respectively. Though the beam quality is slightly deteriorated in the far field, it can still be obviously confirmed that $TE_0 - TE_2$ modes were generated on chip.

Figure R1 | Fabrication of the mode multiplexer. (a) The top-view microscope image of the mode multiplexer. **(b)** The SEM image of the fabricated mode multiplexer. **(c)** The zoom-in SEM image of the waveguide.

Figure R2 | Schematic diagram of the experimental setup for characterizing the mode multiplexer supporting mode conversion.

Figure R3 | The measured far-field of output when the light wave is input from I1-I3, respectively. The measured far-field profile for (a) TE_0 from I1 – O, (b) TE_1 from I2 – O, (c) TE_2 from I3 - O.

3. The crosstalk of the bend was limited to about \sim -15dB (Fig. 2(d)). That is not very low. Usual telecom requirements demand $<$ -30dB. What are some ways to reduce the cross-talk?

Reply: We thank the reviewer for pointing out this weakness. We agree that inter-mode crosstalk is a very important figure of merit (FOM) which should be kept at a low level to maintain the signal to noise ratio (SNR). Ideally, the straight interconnection of two multimode waveguides without defects or bending can be considered to be adiabatic without mode conversion. While the broken symmetry in the bending region causes mode conversion, the digital meta-structure is a highly inhomogeneous media which can be able to compensate that. It is thus possible to figure out a proper device pattern (i.e. the nanohole distribution) with low crosstalk. By doing this, we set the inter-mode crosstalk ($<$ -20 dB) as another optimization objective to re-optimize the digital pattern of the bend. Upon the condition of convergence is met, the simulated inter-mode crosstalk of the bend for the propagation of each mode can be reduced to below -20 dB ($<$ -25 dB in a certain waveband) without degradation of the transmission efficiency. Further improvement might be possible by setting more aggressive objectives, but perhaps at the cost of computational expenses and compromise of other FOMs.

In our previous paper (X. Wu, C. Huang, K. Xu, et. al., J. Lightwave Technol.vol. 36, no. 2, pp. 318-324, 2018), we have performed the theoretical estimation of inter-mode crosstalk based on the approach demonstrated in literature (T. Hayashi, T. Sasaki, and E. Sasaoka, IEICE Trans. Commun., vol. 97, no. 5, pp. 936–944, 2014.). The MDM circuits have measured crosstalk of $<$ -20 dB which will induce \sim 0.5 dB OSNR penalty for 16 QAM and \sim 1dB OSNR penalty for 32 QAM at BER of 1×10^{-3} . There is another recent work also addressing the mode crosstalk issue for high-speed PAM-4 signals. (Y. Hsu, C. Chuang, X. Wu, et al, Photon. Technol. Lett., vol. 30, no. 11, pp. 1052-1055, 2018). The results also indicate that the mode cross talk of $<$ -20 dB is sufficient for negligible signal penalty.

Revision:

[See **Page 2, Paragraph 1-2**]

The descriptions are changed into:

“The simulated insertion loss (ILs) are 0.4 dB, 0.7 dB, 0.84 dB for TE_0 , TE_1 and TE_2 mode, respectively. The crosstalk (CTs) are lower than -20 dB for all the modes over the 80 nm wavelength range, which is sufficient for negligible SNR penalties^{2,4}.”

“The bend is measured to have ILs of 0.71 dB, 0.74 dB, 0.95 dB for TE_0 , TE_1 and TE_2 mode, respectively. The measured CTs are lower than -20dB for all the guiding modes from 1500-1580 nm which is quite consistent with the simulation results.”

Figure 1 | Design and simulation results of the DMS-based bending. (a) Schematic diagram of the designed 3-mode bending with DMS. (b) - (d) The simulated optical field distributions for the micro-bend with three modes (TE_0 - TE_2). (e) - (g) The simulated transmission spectra for the micro-bend with three modes (TE_0 - TE_2).

Figure 2 | Fabrication and experimental results of the DMS-based bending. (a) The top-view microscope image of the MDM circuit consisting of MUX, bending and DEMUX. (b) The microscope image of the reference circuit: back to back (de)MUXs. (c) The SEM image of the fabricated bending. (d) - (f) The measured transmission spectra of the MDM circuit for the optical wave launched from I1 (TE_0 mode), I2 (TE_1 mode), I3 (TE_2 mode), respectively.

4. What is the fabrication tolerance of the devices? How sensitive is the device performance to the size and position of the holes?

Reply: We thank the reviewer's comment on fabrication tolerance. There are many reasons to cause the dimension variations of the nanohole, such as improper dose of exposure, development condition, etching rate and so on. We have performed numerical simulations to study the device tolerance to hole size variations and included the results in the revised Supplement Information as

Note3. The bending has nearly ± 10 nm tolerance to the hole dimension variation with negligible efficiency drop. The crossing and (de)MUX have excellent tolerance to the lowest two modes, but the efficiency for TE_2 mode is very sensitive to the hole dimension variation.

The hole position error mainly results from the positioning accuracy of the electron gun in our experiment. Based on the optimized device pattern, we implement a randomly generated position error to each nanohole. According to the specification of the E-beam writer, the position error ranges with ± 7 nm. The device performances are re-inspected via numerical simulations. We found that all the devices (bending, crossing and MUX) can well tolerate the hole position errors without performance degradation.

Revision:

[See **Supplementary Note3**]:

Supplementary Note3 | Numerical Analysis of the fabrication tolerance

“The ultra-compact, highly functional and efficient devices are mainly realized by the sophisticatedly engineered distribution of the nanoholes. There are many reasons to cause the variations of the nanoholes in fabrication process, such as improper dose of exposure, development condition, etching rate and so on. Here we numerically analyze the device tolerance to fabrication errors in pixel dimension, position and geometry.”

1. The fabrication tolerance of hole dimension

The simulated transmission spectra of the bending under ± 20 nm pixel size variations from 1500 nm to 1580 nm for TE_0 , TE_1 and TE_2 are shown in Supplementary Figure 3 (a) - (c), respectively. The simulated crosstalk performances of the bending are shown in Supplementary Figure 3 (d) - (i). The results indicate that the ILs for each mode can roughly tolerate -20 to +10 nm pixel size variation. The CTs have lower tolerance than ILs, but the CTs can still be lower than -15 dB for all the cases. Supplementary Figure 4 shows the simulated ILs and CTs of the waveguide crossing under ± 20 nm pixel size variations. It can be seen that the ILs of the crossing have no significant change within ± 20 nm variations for TE_0 and TE_1 , but the transmission performance is very sensitive to pixel size variation for TE_2 mode. With ± 20 nm pixel size variations, the CTs can be lower than -20 dB for all the cases. Supplementary Figure 5 shows the simulated ILs and CTs of the (de)MUX under ± 20 nm pixel size variations. The ILs of the (de)MUX has very low tolerance since the mode-convert efficiency is very sensitive to the index profile of the multimode region. The CTs of the (de)MUX device can tolerate the hole dimension variation from -10 nm to +20 nm.

Supplementary Figure 3 | The simulated transmission spectra of the micro-bending under ± 20 nm pixel size variations. (a) - (c) The simulated ILs for (a) $TE_0 - TE_0$ (b) $TE_1 - TE_1$ (c) $TE_2 - TE_2$. (d) - (i) The simulated CTs for three different modes.

Supplementary Figure 4 | The simulated transmission spectra of the 3-mode crossing under ± 20 nm pixel size variations. (a) - (c) The simulated ILs for (a) $TE_0 - TE_0$ (b) $TE_1 - TE_1$ (c) $TE_2 - TE_2$. (d) - (i) The simulated CTs for three different modes.

Supplementary Figure 5 | The simulated transmission spectra of the mode (de)multiplexer under ± 20 nm pixel size variations. (a) - (c) The simulated ILs for (a) $TE_0 - TE_0$ (b) $TE_0 - TE_1$ (c) $TE_0 - TE_2$. (d) - (i) The simulated mode-convert CTs for three different modes.

2. The fabrication tolerance of hole position error.

The hole position error mainly results from the positioning accuracy of the electron gun in our experiment. According to the specification of the E-beam writer, the positioning error is within ± 7 nm. As illustrated in Supplementary Figure 6, the center position offset of the nanohole can be in an arbitrary direction. Based on the optimized device pattern, a randomly generated position error is implemented to each nanohole, which will rearrange the overall pattern as shown in Supplementary Figure 6 (b). Then, we select four randomly generated nanohole distributions (defined as Pattern 1, 2, 3, 4), and simulate the device performance. For the waveguide bending, the simulated ILs for different nanohole distributions are shown in Supplementary Figure 7 (a), (b), (c) for TE_0 , TE_1 and TE_2 mode, respectively. The CT performances are also simulated and shown in Supplementary Figure 7 (d)-(i). We found that the device can well tolerate the position error of the nanoholes. We have done similar analysis for the waveguide crossing and (de)MUX, as shown in Supplementary Figure 8 and 9, respectively. The results show that the crossing and MUX devices are tolerant to the nanohole position error of ± 7 nm as well.

Supplementary Figure 6 | The schematic illustration of hole randomly generated nanohole position error. (a) The illustration of the nanohole position error within ± 7 nm in a random direction. (b) The schematic diagram of the device with randomly generated position errors.

Supplementary Figure 7 | The simulated transmission spectra of the bending with a randomly generated nanohole position errors within 7 nm. (a) - (c) The simulated ILs for (a) $TE_0 - TE_0$ (b) $TE_1 - TE_1$ (c) $TE_2 - TE_2$. (d) - (i) The simulated CTs for three different modes.

Supplementary Figure 8 | The simulated transmission spectra of the 3-mode crossing with a randomly generated nanohole position errors within 7 nm. (a) - (c) The simulated ILs for (a) $TE_0 - TE_0$ (b) $TE_1 - TE_1$ (c) $TE_2 - TE_2$. (d) - (i) The simulated CTs for three different modes.

Supplementary Figure 9 | The simulated transmission spectra of the mode (de)multiplexer with a randomly generated nanohole position errors within 7 nm. (a) - (c) The simulated mode-convert ILs for (a) TE_0 - TE_0 (b) TE_0 - TE_1 (c) TE_0 - TE_2 . (d) - (i) The simulated CTs for three different modes.

Reviewer #2 (Remarks to the Author):

The paper summarizes results of an interesting inverse design approach to realizing key passive components for routing 3-mode waveguides in Si-phonic integrated circuits.

The paper is generally well written and might become suitable for publication provided a few key comments are addressed:

1. The paper is generally well structured but would benefit a lot from the addition of a table summarizing the performance (loss, cross talk etc.) of the different components.

Reply: We thank the reviewer for the useful suggestion. We summarize the performances of the mode MUX, bending and crossing in a table and include the table in the Supplementary Information as Note2.

Revision:

[See **Supplementary Note2**]:

In page 5, the 1st paragraph of Discussion part: “The performances of the devices are summarized in Supplementary Note2.”

Supplementary Note2 | Summary of device performance

Here we summarize the measured transmission performance the different components (3-mode bending, crossing, MUX), MDM circuit 1 and MDM circuit 2 in Supplementary Table 1. It can be seen that all ILs of the different components are less than 1 dB, which is consistent with the design expectations and the average CTs from 1500 - 1580 nm are lower than -25 dB, which can satisfy the device requirements demand. In addition, the ILs of MDM1 and MDM2 are consistent with expectations and CTs are low (< -34 dB).

Device	ILs (dB)			CTs (dB)					
	TE ₀	TE ₁	TE ₂	TE ₀ -TE ₁	TE ₀ -TE ₂	TE ₁ -TE ₀	TE ₁ -TE ₂	TE ₂ -TE ₀	TE ₂ -TE ₁
3-mode bending	0.71	0.74	0.95	-25.01	-31.00	-26.21	-28.24	-31.98	-26.67
3-mode crossing	0.28	0.68	0.92	-26.80	-32.16	-26.54	-29.44	-33.26	-28.74
3-mode MUX	0.68	0.91	0.92	-24.27	-27.72	-23.84	-28.37	-29.65	-27.12
MDM1	7.36	8.22	8.65	-34.75	-36.19	-34.17	-36.91	-36.12	-35.56
MDM2	10.67	11.67	13.12	-39.03	-39.01	-41.17	-33.75	-35.28	-36.54

Supplementary Table 1 | the measured ILs and CTs of the devices from 1500 – 1580 nm.

2. The use of the DMT format to characterize the devices seems quite unfortunate. As I understand the explanation in the paper, the bit loading is optimized in each of the cases (B2B, MDM1, MDM2). This makes sense if the purpose was to maximize the amount of transmitted information. However, given that the purpose here is to characterize the devices it would have been far better to use exactly the same signal in all cases. As it is now one needs to ask the question: why does the optimum bit loading change? You have almost the same SNR in all cases, so what is happening in the device to cause this change?

Reply: The insightful comment and suggestion is highly appreciated. The reviewer is correct that it is better to use the same signal for characterizing the devices rather than the optimum but slightly different bit-loading signals. It is true the SNR responses are similar for all the cases, but slight difference can also be observed, and thus optimum bit loading changes according to Fischer algorithm. This is mainly due to the differences between the two circuits in terms of insertion loss and cross talk. In the revision, we have re-designed and fabricated the devices with improved cross talk. The circuits (MDM1&2) are then characterized by the same DMT signals at both 100 Gbit/s and 112 Gbit/s with fixed bit loading optimized with respect to the B2B case. The reduced insertion loss and improved cross talk are the main reasons for successfully achieving three-channel 112-Gbit/s MDM routing. To further characterize the performance of the devices, the same PAM-4 signals (without bit loading) at 80 Gbit/s have also been transmitted, and the results are included in the Supplementary Information Note4.

Revision:

[See **Manuscript**]:

Page 4, paragraph 2:

In addition to simply measuring the circuit loss and cross talk, we verify the capability of high-speed signal routing via the proposed MDM circuits. Each stream of 112 Gbit/s DMT signals with identical bit-allocation are transmitted through each channel of the MDM circuits one by one. The detail experimental setup and the digital signal processing technique can be found in the Supplementary Note4. The measured signal-to-noise ratio (SNR) responses at the back-to-back (B2B) case and MDM transmission are plotted in the Figure 7. The SNR responses after the MDM circuits transmission have very slight difference compared with the B2B case. The slight differences imply the small variations among different channels due to the insertion loss, crosstalk, and even the measurement instability. The SNR response measured at the B2B case is used to generate a DMT signal by bit-allocation algorithm, shown in Figure 7. The maximum bit-allocation is 5 which corresponds to 32-QAM. The same DMT signals are then routed in MDM Circuit 1 and 2.

Figure 7 | Experimental results of the high-speed signal routing in MDM circuits. (a) Bit allocation of the 112 Gbit/s DMT: the SNR response for B2B and MDM circuits transmission, and the bit allocation for different subcarriers. Inset: constellations of different subcarriers. (b) Measured BER curves for the B2B and MDM transmission of 100 Gbit/s and 112 Gbit/s DMT.

Page 4, paragraph 3:

Besides, the characterization has also been performed at lower speed of 80 Gbit/s 4-level pulse-amplitude-modulation (PAM-4) (See Supplementary Note4).

[See **Supplementary Note4**]: Page 8:

“Erbium doped fiber amplifiers (EDFA) are used before and after the chip for compensating the chip loss, and an optical bandpass filter (OBF) is employed to reduce the amplified spontaneous emission noise after two-state EDFA. The spectrum after OBF is shown in the inset of Supplementary Figure 14.

The DMT signal has a data rate of 112 Gbit/s, with 160 sub-carriers within a bandwidth of 32 GHz.

The optimized bit allocation is fixed for the back-to-back (B2B) case so that the same DMT signals are transmitted through the MDM circuits. Thus, the devices can be characterized through the high-speed transmission of the DMT signal.”

[See **Supplementary Note4**]: Page 9:

“The characterization has also been performed at lower speed of 100 Gbit/s DMT and 80 Gbit/s 4-level pulse-amplitude-modulation (PAM-4). The constellations of sub-carrier (QAM16/32) for 100 Gbit/s and 112 Gbit/s DMT signal under back-to-back, MDM circuit1 transmission, and MDM circuit2 transmission scenarios are shown in Supplementary Figure 15. The constellations of the The slightly degradation of the constellation is due to the decreased SNR induced by the chip insertion loss and inter-modal cross talk. The results well agree with the BER as shown in Fig.7 (b) in the manuscript.

Supplementary Figure 15 | Constellation diagrams of the QAM16 (carrier-69) and QAM32 (carrier-9) sub-carrier.

Supplementary Figure 16 | Measured BER curves for B2B and MDM transmission of 80Gbit/s PAM-4. Inset: The eye diagram of the PAM-4 signals for B2B case and after transmission via MDM2-CH2.

The identical pre-distorted PAM-4 signal (without frequency domain bit allocation) at 80-Gbit/s is transmitted in the MDM circuits in order to characterize the devices’ performance fairly. The BER curves under different received optical power are plotted in Supplementary Figure 16, and eye

diagrams of B2B and MDM circuit2 transmission scenarios are depicted in the insets. The BERs are well below the 20% FEC limit. We can see from the eye diagrams that MDM transmission does not bring in obvious degerenarion."

3. Please state the exact bitrate subtracting the overhead due to FEC. Given that the bit loading changes, it would seem that the bitrate is different in the three cases.

Reply: Thanks for the suggestions. We have fixed the bit loading according to the B2B case so that the same DMT signal is utilized for the characterization for all cases. Thus, the raw bitrate is also fixed. The net bitrate (subtracting 20% FEC overhead) in the revision is 93.34 Gbit/s (for 112 Gbit/s). In practice, the demanded overhead will change depending on the BER and the FEC coding method, and thus the calculated net bitrate will also change accordingly.

Revision:

[See **Supplementary Note4**]: Page 8:

The raw bitrate of 112 Gbit/s is calculated by $517\text{bits}/330 \times 64\text{GSa/s} = 112\text{Gbit/s}$, where 517 is the total bit number of one DMT symbol, 330 represents the point number of one DMT symbol which contains 160 sub-carriers and 10 cyclic prefix, and one DMT symbol maintains 330/64 ns. The calculated bit error rates (BER) under different received power for the DMT signals are shown in Figure 8 in the manuscript, with BER curves well below 20% FEC limit. Given 20% overhead FEC for error-free threshold, the net bitrate is then 93.34 Gbit/s.

4. As I read the description in the supplementary material, it seems like only one modulated data signal was passed through the device when characterizing it. Is this the case? If so, why does the paper refer to 3x100 Gbit/s? This should be clarified/corrected.

Reply: Thanks for the comments. Indeed, the three channels of signals were not transmitted simultaneously. Due to some experimental constraints, the on-chip transmissions of high-speed signals (encoded on different modes) were demonstrated one by one. The concern on the signal qualities for simultaneous transmission of three channels of data is mainly dependent on the inter-mode crosstalk. We have addressed this point in our previous publication (X. Wu, C. Huang, K. Xu, et. al., J. Lightwave Technol.vol. 36, no. 2, pp. 318-324, 2018). We have performed the theoretical estimation of inter-mode crosstalk based on the approach demonstrated in literature (T. Hayashi, T. Sasaki, and E. Sasaoka, IEICE Trans. Commun., vol. 97, no. 5, pp. 936-944, 2014.). The MDM circuits have measured crosstalk of < -20 dB which will induce ~ 0.5 dB OSNR penalty for 16 QAM and ~1dB OSNR penalty for 32 QAM at BER of 1×10^{-3} . There is another recent work also addressing the mode crosstalk issue for high-speed PAM-4 signals. (Y. Hsu, C. Chuang, X. Wu, et al, Photon. Technol. Lett., vol. 30, no. 11, pp. 1052-1055, 2018). The results also indicate that the mode cross talk of < -20 dB is sufficient for negligible signal penalty. Thus, we believe that the aggregate data rate of 3x100 Gbit/s can be achieved by simultaneous transmission of MDM signals with acceptable OSNR penalties. To avoid the confusion, we have revised the manuscript to clarify this point.

Revision:

Abstract: Three channels of 112 Gbit/s signals encoded on each mode are arbitrarily routed through the circuits consists of many sharp bends and compact crossing with a bit error rate under forward error correction limit.

Page 1, paragraph3: Finally, 112 Gbit/s signals encoded on each mode are successfully routed along the arbitrarily designed circuits with a bit error rate (BER) under forward error correction (FEC) limit.

Page3, paragraph2: Here, we demonstrate the possibility of arbitrary and compact on-chip routing of three modes of high-speed signals (112 Gbit/s) in a multimode waveguide.

Page5, paragraph2: Thus, the ultra-compact signal routing of three channels of 112 Gbit/s signals is successfully achieved.

5. It is very relevant that the tolerance to fabrication error on the hole size in the structure is characterized, however, what about errors in placement and variation in shape? I think the authors should comment on this in the paper and justify why size variation is the most sensitive to fabrication errors.

Reply: We thank the reviewer for calling our attention to more details of fabrication tolerance. In the revision, we numerically study the device performances with fabrication errors in nanohole size, position, and shape. The revisions for fabrication tolerance to nanohole size and position can be referred to our reply to Question 4 raised by the first reviewer. The study on device tolerance to hole shape has been included in Supplementary Note3 as well.

Revision:

[See **Supplementary Note3**]:

"3. The fabrication tolerance of the hole shape

The shape of nanohole can be distorted elliptically due to the write field misalignment and deterioration of focus. However, this effect can be well negligible if the E-beam is well optimized. In fact, the nanohole shape will be more easily affected by the sidewall roughness induced by the fabrication imperfection. The a randomly generated error is implemented to the perimeter of each nanohole to simulate the sidewall roughness. We consider the error range from -10 nm to 10 nm. Supplementary Figure 10 describes how we simulate the sidewall roughness. To study the impact of the roughness on the device performance, we randomly generated four patterns with different roughness distributions (defined as Pattern 1, 2, 3, 4). For the waveguide bending, the simulated ILS for nanoholes with different roughness distributions are shown in Supplementary Figure 11 (a), (b), (c) for TE₀, TE₁ and TE₂ mode, respectively. It can be seen that the roughness has negligible impact on the transmission efficiency. The CT performances are also simulated and shown in Supplementary Figure 11 (d)-(i). We found that the device can well tolerate the fabrication error of the nanohole shapes. We have performed similar analysis for the waveguide crossing and (de)MUX, as shown in Supplementary Figure 12 and 13, respectively. The results show that the crossing and MUX devices are quite tolerant to the nanohole shape error as well."

Supplementary Figure 10 | The Schematic illustration of the error in shape of the nanohole. (a) Each nanohole has a random roughness within ± 10 nm. **(b)** The schematic diagram of the device with randomly generated

roughness.

Supplementary Figure 11 | The simulated transmission spectra of the bending with randomly generated roughness. (a) - (c) The simulated ILs for (a) TE_0-TE_0 (b) TE_1-TE_1 (c) TE_2-TE_2 . (d) - (i) The simulated CTs for three different modes.

Supplementary Figure 12 | The simulated transmission spectra of the 3-mode crossing with randomly generated roughness. (a) - (c) The simulated ILs for (a) TE_0-TE_0 (b) TE_1-TE_1 (c) TE_2-TE_2 . (d) - (i) The simulated CTs for three different modes.

Supplementary Figure 13 | The simulated transmission spectra of the mode (de)multiplexer with randomly generated roughness. (a) - (c) The simulated mode-convert ILs for (a) TE_0-TE_0 (b) TE_0-TE_1 (c) TE_0-TE_2 . (d) - (i) The simulated CTs for three different modes.

6. Other than causing loss, fabrication errors might also cause increased crosstalk. In actual MDM communication this could be a much more severe limitation than a slight increase in loss. I think the authors should investigate the relevant fabrication tolerances, also in terms of increased crosstalk.

Reply: We thank the reviewer for the constructive suggestions. We have numerically analyzed the crosstalk performance under various fabrication errors such as dimension, position and shape of the nanoholes.

Revision: The revisions for fabrication tolerance study on the crosstalk have been included in Supplementary Note3 as we shown in the replies for previous questions.

Reviewer #3 (Remarks to the Author):

In an “Arbitrarily routed mode-division multiplexed photonic circuits for dense integration” Yingjie Liu et al. have presented a collection of basic (inverse designed) building blocks for use in on-chip mode division multiplexing photonic circuits. This approach presents major advantages over traditional wavelength division multiplexing, and will likely continue to grow in popularity as photonic optimization techniques mature. Although conceptually unremarkable in light of previous publications, the demonstration provided by the authors is thoroughly convincing and offers clear performance advantages over any existing art. With minor revisions to improve the quality of the text, it is entirely suitable for publication in Nature Communications.

Comments:

1. Some additional details about the direct binary search should be included. What does an iteration refer to in this context? What are the convergence conditions for each structure? The figures of merit?

Reply: We thank the reviewer for the positive recommendation for publication in Nature Communications. In the revision, we have added a detail description of how we optimize the device structure via direct binary search (DBS) algorithm jointly with the 3D FDTD.

Revision:

[See **Methods**]:

DBS method:

The design area of the device is discretized into a number of pixels. The geometry of the pixel can be either circle, square, or any other shape. The minimum feature size can be determined according to the fabrication capability. Here, each pixel has a binary state of the material property: silicon or air. The DBS method is used to find a proper distribution of pixels with different material property that satisfied the design targets. The optimization began with an all silicon structure for initialization. Then, the material state of the pixel is toggled one by one, and the figure-of-merits (FOMs) are investigated. Here, the FOMs are ILs and CTs for TE₀-TE₂ modes which are calculated by 3D FDTD with 30nm×30nm×30nm grid size. If the FOMs improve, the toggled pixel state will be saved. The improvement can be verified by the following condition:

$$\left[\frac{\sum_j \eta_{j,i+1}}{\sum_j \eta_{j,i}} \geq 1 \right] \bigcap_{j=2}^3 \left[\left| \frac{\eta_{j,i}}{\eta_{1,i}} - 1 \right| \leq \sigma_{j,i} \right]$$

AND

$$\left[\frac{\sum_k \beta_{k,i+1}}{\sum_k \beta_{k,i}} \leq 1 \right] \bigcap_{k=2}^6 \left[\left| \frac{\beta_{k,i}}{\beta_{1,i}} - 1 \right| \leq \rho_{k,i} \right]$$

where $\eta_{j,i}$ is the transmission efficiency of j-th mode from 1500-1580 nm in the i-th iteration (j=1 for TE₀, j=2 for TE₁ and j=3 for TE₂), and $\beta_{k,i}$ is the k-th CTs from 1500-1580 nm in i-th iteration (k=1 for TE₀ - TE₁, k=2 for TE₀ - TE₂, k=3 for TE₁ - TE₀, k=4 for TE₁ - TE₂, k=5 for TE₂ - TE₀, and k=6 for TE₂ - TE₁). $\sigma_{j,i}$ is the radius of convergence for efficiency of the j-th mode in the i-th iteration, and $\rho_{k,i}$ is the radius of convergence for the k-th CT in the i-th iteration which is a monotone decreasing function.

The first iteration ended after all the pixels are inspected. Then, this process went over again to further improve the FOMs towards to design objectives. Normally, several iterations are needed to meet the condition of convergence where σ and ρ converge to zero [37]:

$$\sigma_{j,i} = \varepsilon \times \left(1 - \frac{\eta_{j,i}}{\eta_{j,\text{objective}}} \right)$$

AND

$$\rho_{k,i} = \delta \times \left(\frac{\beta_{k,i}}{\beta_{k,\text{objective}}} - 1 \right)$$

where ε and δ are the convergence factor, $\eta_{j,\text{objective}}$ is the transmission objective of the j-th mode, $\beta_{k,\text{objective}}$ is the transmission objective of the k-th CT.

The whole design process is performed via an 8-core server. For the bending device, it takes ~ 50 hours in average to get the convergent results after 4 iterations. For the crossing device, it takes ~ 40 hours in average to get the convergent results after 3 iterations.

2. The text has numerous small grammatical errors that need to be corrected. For instance:

The adiabatic wave propagation requires too large footprint for a bent waveguide.

The waveguide width is chosen to be 2.3 μm which can support three lowest order modes and allow for enough number of pixels as well.

DBS method is easy to implement and fast to convergence for the optimization of digital meta-structure with binary material state.

Reply: We appreciate for the reviewer's carefulness in pointing out the grammatical errors.

Revision:

Page1, paragraph1: "To achieve adiabatic wave propagation in a bent waveguide, the required footprint is too large."

Page2, paragraph1: "The waveguide width is chosen to be 2.3 μm , which allows for enough number of pixels."

Page6, paragraph3: "DBS method is easy to implement and fast to convergence for the optimization of digital meta-structure with binary material state." This sentence has been deleted.

We have also carefully revised other grammatical errors in the manuscript.

Reviewers' comments:

Reviewer #1 (Remarks to the Author):

The authors have adequately addressed my comments from the last review.

Reviewer #2 (Remarks to the Author):

The authors have carefully addressed the comments and provided satisfactory responses and revisions to all but one.

I am still worried about the way the demonstrated data rate is presented. I think it is still very easy for a reader to get the mistaken impression that 112 Gbit/s have been simultaneously transmitted on the three modes.

I would strongly recommend that the authors avoid mentioning 'three channels of 112 Gbit/s' anywhere. Similarly in the supplementary material, writing '3x112Gbit/s' can only serve to give the false impression that three channels of 112Gbit/s have been demonstrated in the chip – which, as far as I understand, it has not.

I tend to believe the authors claim that the crosstalk investigation would indicate that the chip could potentially support three channels at 112Gbit/s – but this is not demonstrated and should not be claimed as a result.

If the authors want to claim 3x112Gbit/s then the signal could be split in three before the chip, de-correlated using (fiber) delay lines and then launched into all three inputs on the chip. This would properly emulate a true 3x112Gbit/s scenario.

This is my only remaining reservation before paper would be suitable for publication.

Reviewer #3 (Remarks to the Author):

The authors have addressed all my concerns and in my opinion the paper is should proceed to publication.

Reviewer #1 (Comments to the Author):

The authors have adequately addressed my comments from the last review.

Reply: We thank the reviewer for the 2nd round review process.

Reviewer #2 (Remarks to the Author):

The authors have carefully addressed the comments and provided satisfactory responses and revisions to all but one.

I am still worried about the way the demonstrated data rate is presented. I think it is still very easy for a reader to get the mistaken impression that 112 Gbit/s have been simultaneously transmitted on the three modes. I would strongly recommend that the authors avoid mentioning 'three channels of 112 Gbit/s' anywhere. Similarly, in the supplementary material, writing '3x112Gbit/s' can only serve to give the false impression that three channels of 112Gbit/s have been demonstrated in the chip-which, as far as I understand, it has not.

I tend to believe the authors claim that the crosstalk investigation would indicate that the chip could potentially support three channels at 112Gbit/s - but this is not demonstrated and should not be claimed as a result. If the authors want to claim 3x112Gbit/s then the signal could be split in three before the chip, de-correlated using (fiber) delay lines and then launched into all three inputs on the chip. This would properly emulate a true 3x112Gbit/s scenario.

This is my only remaining reservation before paper would be suitable for publication.

Reply: We appreciate the reviewer's efforts for reconsideration of the manuscript. We completely agree with the reviewer that all the statements of "3x112Gbit/s" should be avoided, since we measure each individual signal transmission separately. We clarify this point in the revised manuscript and supplementary information.

For lower than -20dB CT of the circuits, the additional penalties should be negligible according to the previous theoretical predictions. As suggested by the reviewer, this is not experimentally demonstrated, and we could only expect the potential for simultaneous transmission of MDM signals. Thus, we carefully discuss this point in the revised manuscript according to the reviewer's suggestion.

Revision:

All the claims of "3x112 Gbit/s" or "three channels of..." in both manuscript and supplementary information have been revised to avoid any confusion.

We also explicitly clarify that each individual mode channel is tested separately.

In page 5, the 2nd paragraph: "Though we measure each mode channel of signal transmission separately, we would expect the potential of simultaneous transmission of the mode-multiplexed signals. This is because the additional signal penalties mainly come from the inter mode CT of the

MDM circuits. According to the theoretical predictions^{2,4}, the additional penalties induced by CT would be insignificant for the demonstrated routing circuits."

Reviewer #3 (Remarks to the Author):

The authors have addressed all my concerns and in my opinion the paper is should proceed to publication.

Reply: We thank the reviewer for the positive recommendation.